# Relationship between Milk Yield and Reproductive Parameters on Three Hungarian Dairy Farms

**DOI:** 10.3390/vetsci11050218

**Published:** 2024-05-14

**Authors:** Zsófia Amma, Jenő Reiczigel, Hedvig Fébel, László Solti

**Affiliations:** 1Department of Obstetrics and Farm Animal Medicine Clinic, University of Veterinary Medicine, 1078 Budapest, Hungary; ammazsofi@gmail.com (Z.A.); sol0248@univet.hu (L.S.); 2Department of Biostatistics, University of Veterinary Medicine, 1078 Budapest, Hungary; reiczigel.jeno@univet.hu

**Keywords:** cattle, fertility, high milk production, insemination index, service period

## Abstract

**Simple Summary:**

The remarkable increase in milk yield over the past few decades has been accompanied by the reduced calving to conception interval of dairy cows. The hypothesis of an antagonism between milk yield and reproduction, however, is neither completely understood nor generally accepted. In fact, there is no homogenous relationship between milk yield and fertility. As there are contradictory hypotheses published in the literature, we decided to investigate the relationship between milk production and reproduction parameters on three local Hungarian dairy farms. The data obtained in our experiment indicate that a fertility decrease parallel to increasing milk production does exist.

**Abstract:**

It is postulated that there is negative correlation between milk yield and reproductive performance. However, some studies definitely doubt this causality. The aim of our study was to investigate the relationship between milk production and fertility on three dairy farms. The production parameter was the milk yield (in kg), and fertility was expressed by the number of inseminations per conception (AI index), as well as by the length of the service period (in days). A total of 13 012 lactations from cows with their first three lactations completed were analysed. The number of inseminations was significantly correlated with the milk yield and with the studied farm (*p* < 0.0001), but its correlation with the lactation number was not significant (*p* = 0.9477). A similar relationship was found after evaluating the length of the service period. A multiplicative model showed that a 2000 kg milk increase extended the service period by 9% and increased the AI index by 13%. Thereafter, using quartiles of the cows, the service period of the highest-producing group rose by 41.5 days, and the AI index by almost 1, compared to the lowest quartile. Our results indicate a definitive decline in reproductive indicators parallel to an increase in milk production but did not prove an inevitable correlation.

## 1. Introduction

One of the key factors of economic cattle breeding is their reproduction. Shanks et al. reported that 21% of the direct health costs are attributed to reproductive problems and a further 19% are related to insemination. In other words, approx. 40% of all costs are bound to reproduction [1].

Currently, it is expected that dairy cows produce around or above 12,000 kg of milk annually. According to recent statistics, the global trend of milk yield’s increase is uninterrupted: in the USA, 60 percent more milk is produced from 30 percent fewer cows than in 1967 [2]. This is because each cow produces over 2.5 times as much milk as 50 years ago. There is no reason to believe there will be changes in this trend over the next 50 years (University of Minnesota, 2019).

The United Kingdom statistic is similar: during the period of 2002/03, the average milk yield was 6450 L per cow per year [3]. This value increased in the following years, reaching 8133 L per cow per year in 2022/23.

In the European Union, the apparent milk yield continued to rise in 2021 [4]. The production of raw milk on the EU’s farms was an estimated 161 million tons, which would represent a year-by-year increase of 0.7 million tons.

As early as 1993, Nebel and McGilliard hypothesized that there might be some correlation between reproductive traits and milk yield [5]. Butler demonstrated that between 1951 and 2001 the average milk yield of Holstein Friesian cattle has doubled, whereas their conception rate decreased from about 70% to 30% [6]. The extended calving interval results in a financial loss that has been calculated by several authors. According to these authors, a calving interval elongated by 37 days results in a loss of EUR 2.08 per cow per day: in total, EUR 77 per cow. Every missed pregnancy causes a USD 640 loss [7]. 

The hypothesis of an antagonism between milk yield and reproduction, however, is neither completely understood nor generally accepted. In fact, there is no homogenous relationship between milk yield and fertility. Lucy (2001) demonstrated that the remarkable increase in milk yield over the past few decades has been accompanied by an obvious decrease in the reproductive efficiency of dairy cows [8]. A 16-year study recorded an increase in milk production from 8300 to 11,221 kg (44%), accompanied by a 6.8% lower first insemination conception rate (from 39.1 to 32.3%). Every 1000 kg increase in milk was equivalent to a 2.3% drop in conception rate [9]. A similar trend has been reported in another study, in which the conception rate of cows producing above 8597 kg of milk was 49%, while that of cows producing below 6783 kg was 55%, despite the fact that the management of high-producing animals is generally better. It has been estimated that every 100 kg of milk increase is linked to 1 additional day open [10,11].

However, according to some authors, even if there is a herd-level antagonism between milk yield and fertility, it does not mean necessarily that within the population the highest producing cows suffer from poorer fertility. They found just the contrary: in some research the high-yielding cows showed an even better conception rate and shorter open days. Laben et al. found a one-cycle-earlier conception and better pregnancy rate in the highest yielding group of cows [12]. These statements were confirmed by other authors [5,13,14]. In a study, the interval until the first postpartum insemination of high-yielding cows proved to be shorter by 13 days, and the service period by 16 days, than that of low-producing ones. López-Gatius et al. also reported a positive correlation between the milk yield and fertility parameters. According to them, the chance of conception showed a 6.8 fold increase in cows with a ≥50 L daily milk production, and their calving-to-conception period shortened by 1.8 days [15]. Thus, they doubted that the increasing milk yield alone would necessarily exert an unfavourable effect on reproduction. 

As the contradictory hypotheses found in the literature demonstrate, the dogma of an antagonism between milk yield and reproduction is not generally accepted. These conflicting statements do not allow us to conclude whether there is a necessarily negative correlation between the increasing milk yield and decreasing reproduction rate. The aim of our present study was to investigate the relationship between milk production and fertility on some local dairy farms.

## 2. Materials and Methods

### 2.1. Animals and Management

The experiments were conducted on three large dairy farms (Farm 1, 2 and 3) in southern Hungary: retrospective data were collected from the period between January 2010 and August 2022. The ambient temperature typically changed according to the seasons, ranging from −10 °C in winter to +35 °C in summer. The number of animals fluctuated year by year, between 700 (Farm 1 and 3) and 500 (Farm 2), according to market conditions. Holstein Friesian cows were chosen because they are considered a high-yielding breed. The management level of the examined farms was average, so management errors were minimised. Throughout the study, the milk yield is defined as a cow’s milk production adjusted for 305 days. On the farms, three daily milkings were performed. After calving, both the primiparous heifers and the multiparous cows were placed in the production group. Between Day-14 and Day-21 after calving, healthy cows were sub-grouped according to their milk yield and kept there until their body condition score (BCS) fell <3 or their milk production dropped ≥10 percent below the group’s average. The animals were fed with TMR (Total Mixed Ration) twice daily (27 kg/day dry matter intake), which was pushed up 6–8 times a day. The composition of the mix changed but, in general, contained forages (corn silage, alfalfa haylage), grains (corn, wheat, barley), protein feeds (extracted soybean meal, extracted rapeseed meal), minerals and vitamins. Rations were calculated for about a 20 percent higher nutrient intake than the group’s average milk production.

### 2.2. Examined Farms and Technology

The management conditions of different farms are summarised in Table 1. The differences between the farms in their housing of dairy cows and used protocols regarding reproduction may be compared.

### 2.3. Data Collection

For minimizing the distortion effect caused by unspecific health conditions like lameness, mastitis and other diseases, only the cows with their first three lactations completed were included in the survey. The milk yield was adjusted for 305 days and expressed in kg in order to reflect better the overall lactation performance, rendering our conclusions more stable and independent of changing circumstances. Fertility status was evaluated by the number of inseminations required for conception (AI index), as well as by the length of the service period (days open from calving to conception). Animals with more than one lactation were regarded as separate individuals per lactation, thus a total of 13,012 lactations were studied. The data distribution was the following: 6418 first, 4223 second and 2371 third lactations were included. A total of 5266 data points were analysed from Farm 1, 3339 from Farm 2 and 4407 from Farm 3. The average milk production of the three examined farms was 11,423 kg and the overall length of their service period was 165.7 days.

### 2.4. Statistical Analysis

The relationship between AI index (number of inseminations) and milk production—taking into account the number of lactations and the possible effect of the farm management—was analysed by Poisson regression with the AI index as the dependent variable and milk production, farm, and number of lactations as explanatory variables. The length of the service period was first analysed using a linear model, with the logarithm of the service period as the dependent variable and milk production, farm, and the number of lactations as the explanatory variables. In an additional analysis, animals were divided into four production groups using the sample quartiles of their milk production as the cutpoints. Results with *p* ≤ 0.05 were regarded as significant. The analyses were performed in the R 4.2.1 program.

## 3. Results

A data analysis with a Poisson regression proved that the number of inseminations is significantly correlated with the milk yield and with the studied farm (*p* < 0.0001), but there was no significant correlation with the lactation number (*p* = 0.9477). When evaluating the length of the service period with a linear model, a similar relationship was found. Again, a multiplicative model was used, along with the logarithm of the length of service period. Connected to rising milk production, both the number of inseminations (AI index, Figure 1) and the logarithm of the length of the service period (Days open, Figure 2) clearly increased. We have also calculated how every kg milk increase affects the service period, and expressed this in percentages. The results show that every 1 kg milk increase increased the number of open days by 0.0044%, which means a 9% increase per 2000 kg of milk. The AI index increased by 0.0062% per kilogram, which is about 13% for a 2000 kg increase in milk yield.

The farms were also compared to each other in terms of their service period and of their number of inseminations (Figure 3). According to the results of the multiplicative model, the length of the service period at Farm 1 does not differ notably from that of Farm 3 (0.5% shorter on Farm 1) but differs greatly from that of Farm 2. On a logarithmic scale, this difference is 0.036, which means an approx. 8.7% longer service period. Despite this finding, the AI index on Farm 2 is only 0.6% higher than that on Farm 1. The length of the service period was approximately the same on Farm 1 and Farm 3, but the AI index on Farm 3 was 14.2% higher. 

In our second run, the fertility indicators were examined by allotting the animals into four performance groups based on their milk yield (always using the quartiles of the experimental cows). In that way, each category contained an equal 3253 lactations. These groups, along with their milk production and main fertility indicators, were as follows (Table 2): the lowest production group had an average service period of 146.1 days, while the highest quartile had on of 187.7 days. This means that the increasing milk yield increased the number of open days by 41.6 days.

In line with the above results, the AI index of the highest production group increased from 2.78 to 3.77 compared to the lowest quartile. The AI index of Farm 3 (Figure 4) and the length of the service period of Farm 2 (Figure 5) showed extreme values. Interestingly, however, at Farm 3, between the third and fourth quartile—despite increasing milk production—there were minimal changes in their reproductive indices. Figure 4 shows clearly that the increase in the AI index is not as high as, for example, that between the second and third group. In addition, Figure 5 shows that, in spite of the increasing milk yield, the length of the service period did not change on Farm 3. These results cast doubt on the antagonistic relationship between milk production and fertility and suggest further studies are required.

In the final stage of the study, to avoid any distortion caused by extreme values, these were filtered and removed from the dataset (results not shown in any table). Accordingly, lactations <5000 kg, service periods >200 days or AI indices >7 were excluded. Thereafter, with a total of 9217 lactations, the above examinations were run again. From each of these methods it was found that, after these omissions, the trend remained the same and the results did not change.

## 4. Discussion

When we designed these experiments, we expected evidence that the increasing milk yield would not affect necessarily the fertility of cows, as indicated by some of the data in the literature. In contrast to this preliminary expectation, a firm negative relationship between milk production and reproductive parameters has been found in our study. These findings are in good accordance with the statements of Lucy and Crooker; Washburn et al.; Evans et al.; and many others who all thought that the increasing milk yield negatively affects the reproduction of cows [10,11,16]. At the same time, they contradict the results of others like Laben et al. [12], according to whom this small but real antagonism, with reproductive efficiency, may be overshadowed by good management; effective oestrus detection probably is a major factor. It is also in contrast to López-Gatius et al., who believe that this negative correlation—if exists at all—is not inevitable [15].

Our data indicate a definitive decline in reproductive indicators along with increasing milk production. However, because this decline seems to be environmental rather than genetic, further investigations are needed to dispel the doubts raised about this inevitable antagonism. If it is not genetic and not inevitable, what other factors can be postulated to be at work in the background?

### 4.1. Negative Energy Balance (NEB)

The first candidate is the postpartum NEB, which might be responsible indirectly through inadequate metabolic adaptations. On the ovarian level, the unfavourable effect of a NEB is a change in the follicular fluid’s composition. Britt provided a hypothesis about the impact and latent effect of postpartum losses in body condition on the follicular development, corpus luteum formation and function, and fertility in high-producing dairy cows [17]. The change in follicular fluid composition is reflected by elevated total cholesterol, β-OH butyric acid, urea, NEFA and lower glucose values. These intrafollicular changes are detrimental to the growing oocytes and later to the quality of embryos, whose trophectoderm cells may not be able to produce a sufficient amount of interferon-tau (bIFNτ) for a maternal recognition of pregnancy [18,19]. Poor fertility, however, cannot be explained simply by NEB because it is possible to select concomitantly for both parameters [20,21].

To monitor the metabolic status, regular body condition scoring (BCS) is recommended to achieve an appropriate value at calving. The BCS at which a cow calves, nadir BCS, and the amount of BCS she loses post calving are associated with milk production, reproduction and health [22]. The recommended BCS at parturition is 2.75 to 3.0 and this value must not decrease by more than 0.5 points until the first insemination [23]. According to Carvalho et al., the primary factor affecting conception rate is the cow’s BCS during the early lactation period [24]. If the condition loss during lactation does not exceed 0.5 BCS points, this prevents the manifestation of serious fertility problems, decreased conception rates and an extension of open days [25]. More multiparous cows losing >0.5 units of BCS tended to have greater pregnancy losses, such as second-parity cows and cows of third or greater parity [26]. Increasing the BCS at insemination was associated with decreased pregnancy loss.

In our study, the energy balance was supported by feeding the cows TMR, which met the energy need of the cows. To avoid the development of a serious NEB, an important part of postpartum management is the effort to keep the energy budget as balanced as possible. Therefore, feeding cows an energy-dense ration with fat supplementation or an increased starch content may be beneficial—it has been found that this feeding regime increased the number of cows that displayed heat and ovulated by day 50. However, recent findings suggest that high-energy diets can promote follicular growth in parallel with an increase in plasma insulin concentrations but that they have a detrimental effect on the quality of oocytes [27].

The energy deficit may be aggravated by climate change: the appetite of cows exposed to heat stress is affected and their NEB is even worse [28]. In addition to this, embryonic mortality may also be higher [25,29]. Recently, with in vitro-produced embryos, Nishisozu et al. concluded that an ambient temperature >25 °C and total humidity index >75% adversely affect the conception rate outcome in cows but not in heifers [30].

### 4.2. Weak or Silent Oestrus, Inadequate Heat Detection

In high-yielding cows, the most frequent failure is a missing or silent heat, with a prevalence of 42.1%, followed by ovarian hypofunction (12%) and cystic ovarian disease (6.3%) [9]. This hypothesis is supported by the shorter duration of heat: cows producing 25 to 30 kg of milk daily displayed oestrous behaviour for 14.7 h, in contrast to cows producing 50 to 55 kg of milk, who were showing this behaviour for only 2.8 h [31].

Oestrus detection seems to be quite simple but, in fact, this is one of the weakest points on some large-scale dairy farms. In a Swedish study, the conception rate was found to increase with stronger oestrus intensity, from 24% with weaker and more uncertain heat symptoms, e.g., a red and swollen vulva, to 54% for primary oestrus symptoms like standing heat [32]. Heat detection inaccuracy negatively impacts the pregnancies per AI, as it increases the inseminations per pregnancy with little probability of conception, while also having the potential to disrupt established pregnancies. Almost half of the cows in Ireland displayed mounting activity for ≤8 h, highlighting the importance of conducting 3–4 periods of oestrous observation daily [33].

We tend to accept the hypothesis that weak and short oestrus symptoms may be an important causative factor responsible for delayed conception and a higher AI index. Therefore, an improved heat check augmented by oestrus detection aids like tail paint, pressure detectors, progesterone analyses, monitoring activity by motion sensors, etc., is necessary. In our experiment, in terms of herd management, Farm 1 proved to be the best. On this farm, the cows were not monitored for heat but treated by double Ovsynch and inseminated at fixed times. This coincides with findings of Borchardt et al., where there was a benefit in terms of pregnancy per insemination for cows with a 100% timed artificial insemination after completing a Presynch–Ovsynch protocol [34]. In fact, Ovsynch renders the cow an “induced ovulator”, allowing for a timed insemination without performing heat detection [35]. On Farm 1, rubber flooring was used, which is postulated to affect the earlier onset of regular oestrus behaviour and improve the calving interval through earlier conception and less days open, but does not cause higher milk yields [36].

On Farm 2 and Farm 3, there was a regular heat check and the animals were inseminated when they displayed heat. However, the cows found open at their bi-weekly ultrasound pregnancy check were treated with prostaglandin (Farm 1) or Ovsynch (Farm 3). This corresponds to the findings of Peters and Pursley [37], who used the Ovsynch protocol and found a higher conception rate in cows above the herd average compared to cows with low milk production. Similarly, Gvozdic et al. recommended some type of synchronization—on those farms where oestrus detection does not represent a problem, prostaglandin-based or SelectSynch protocols are preferred [38]. However, on farms with insufficient heat detection, Ovsynch or a combined Presynch + Ovsynch treatment is recommended, preferably started mid-cycle [39]. This is in good agreement with the recommendation of Borchardt et al., who experienced that cows with a 100% timed insemination after completing a Presynch + Ovsynch protocol had a higher conception rate compared to cows that were inseminated after oestrous detection or who received fixed-time insemination showing no signs of oestrus [34]. This protocol was used on Farm 1 but partly also on Farm 2 and 3—where the animals were inseminated at detected heat—on cows found open at their pregnancy examination.

Apart from the mentioned findings, the elevated steroid metabolic clearance rate may play a key role in reproductive failures. Higher feed consumption raises mesenterial blood circulation and thus intensifies the hormone metabolism and elimination of the liver. As a result, the circulating levels of oestrogen and progesterone in lactating cows are lower compared to their non-lactating counterparts [40,41]. A data analysis of dairy cows demonstrated that, in the high-producing group, their milk yield and dry matter intake was higher, whereas their plasma progesterone proved to be lower than that of random control cows. Therefore, the conception rate of cows (25–40%) is generally worse than that of heifers (40–75%) [42]. In our retrospective study, the progesterone level of cows was not monitored for checking their steroid metabolic clearance rate as a causative factor.

### 4.3. Embryonic or Foetal Loss

According to a hypothesis, intensive milking could increase embryonic or foetal death (see above in relation to NEB and body condition scoring). Pregnancy loss may occur at any point of gestation, with the largest percentage of loss occurring in the first 30 days and, subsequently, decreasing as the pregnancy progresses [43]. However, Santos et al. did not find any correlation between the milk yield and embryonic or foetal loss [29]. High-yielding cows are typically well nourished and healthy, which does not inhibit but rather promotes the resumption of first oestrus. Pregnancy loss has not been studied in our experiment.

## 5. Conclusions

The main question in this study was whether there is any correlation between milk yield and reproduction. The data obtained in our experiment indicate that a fertility decrease parallel to increasing milk production does exist. Although these findings show a definitive decline in reproductive indicators, further investigations are needed to elucidate to what extent is this negative relationship causal. To overcome the impaired fertility in high-yielding cows, professional reproductive management is beneficial. However, alone, the fact that the fertility problems can be diminished by high-level management suggests that the antagonism between milk production and fertility is rather environmental than genetic. More experiments are required to dispel the doubts raised against the hypothesis of an inevitable decrease in fertility.

## Figures and Tables

**Figure 1 vetsci-11-00218-f001:**
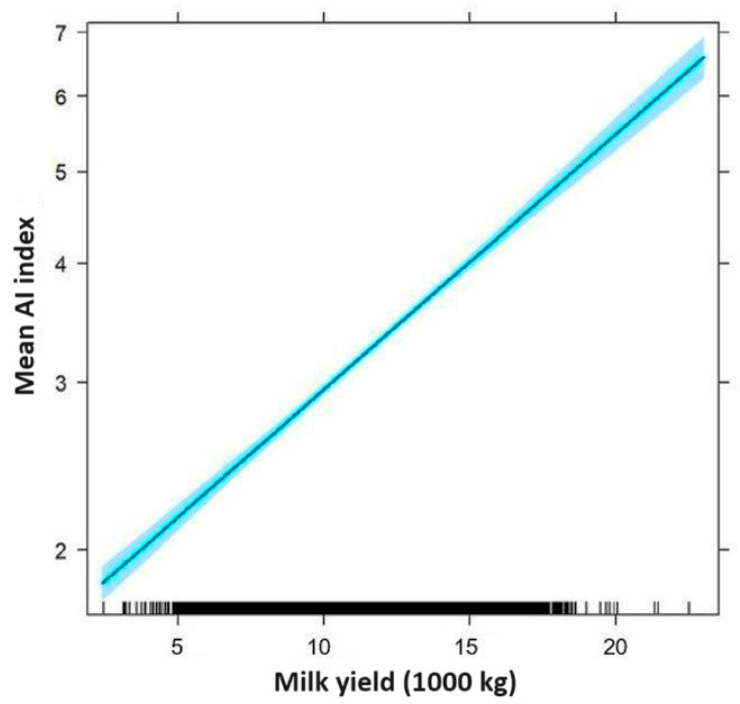
Increase in the AI index in relation to the milk yield (the blue shaded area is the 95% confidence band).

**Figure 2 vetsci-11-00218-f002:**
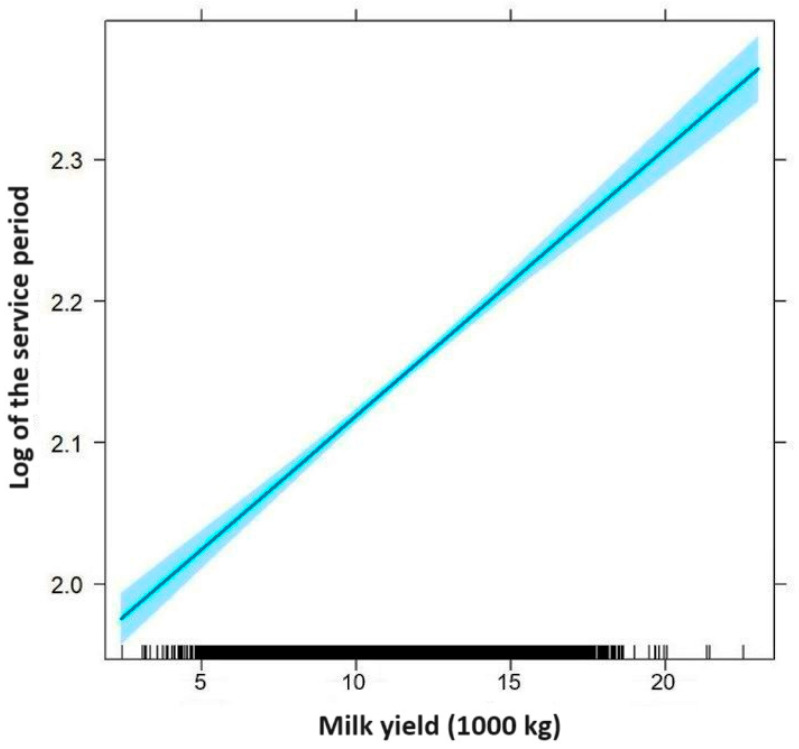
Increase in the service period related to the milk yield (the blue shaded area is the 95% confidence band).

**Figure 3 vetsci-11-00218-f003:**
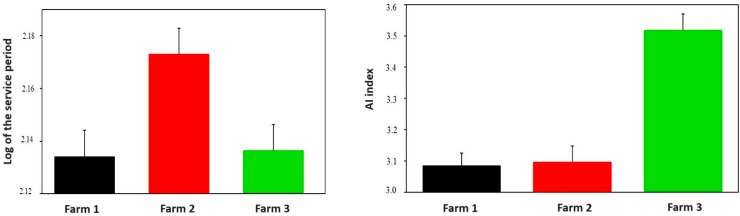
Evolution of the logarithms of service period and of the fertility index of the surveyed farms.

**Figure 4 vetsci-11-00218-f004:**
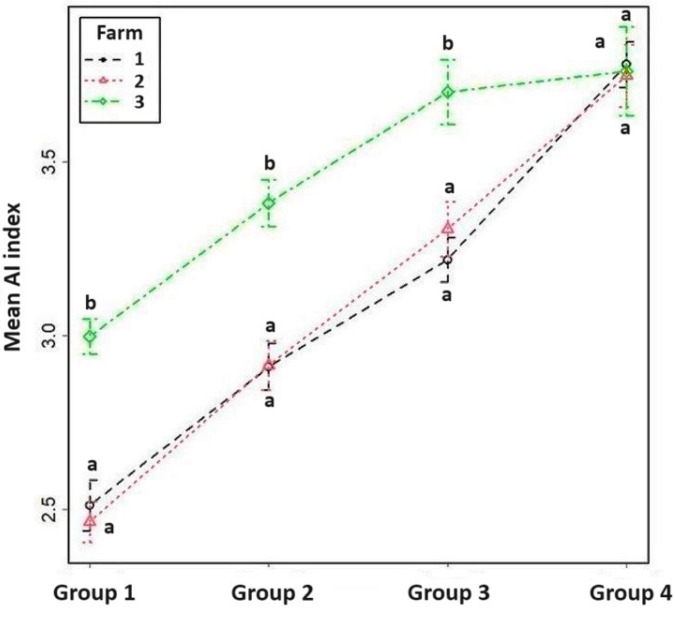
The mean AI index in the four production groups based on milk yield and the surveyed farms. Analysis using a Poisson regression model showed that the effects of both factors, and the interaction between them, are significant. ^a,b^ Means with different superscripts within the same group differ (*p* < 0.05).

**Figure 5 vetsci-11-00218-f005:**
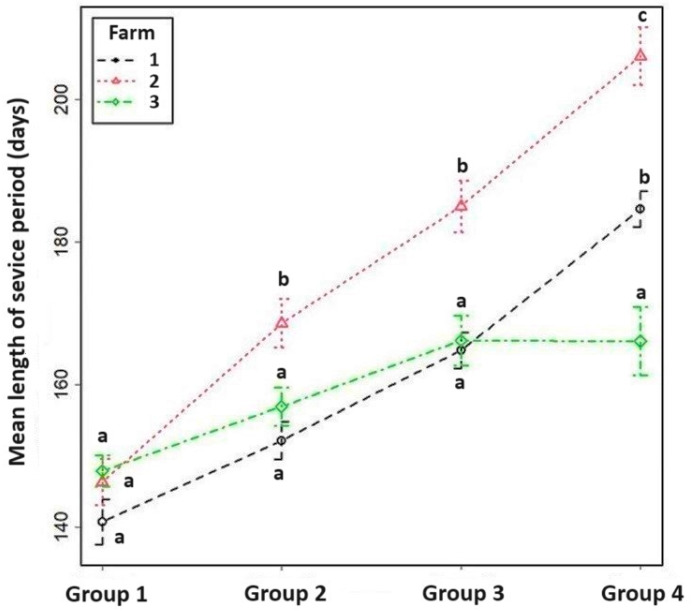
The mean length of the service period of the four production groups on the surveyed farms. Analysis using a linear model showed that the effects of both factors, and the interaction between them, are significant. ^a,b,c^ Means with different superscripts within the same group differ (*p* < 0.05).

**Table 1 vetsci-11-00218-t001:** Management conditions of different farms.

	Farm 1	Farm 2	Farm 3
Flooring	Rubber floor	Deep litter	Deep litter
Temperature control of barns	Cross ventilation, waterspray	Cross ventilation, waterspray	Cross ventilation, waterspray
Heat monitoring	No	Yes	Yes
Oestrus synchronization	Double Ovsynch of all animals	Ovsynch of nonpregnant animals only	Ovsynch of nonpregnant animals only
First AI (0.25 mL frozen–thawed semen)	Fixed time AI at D-80 post partum	At detected heat	At detected heat
Pregnancy check	Ultrasonography D-32	Ultrasonography	Ultrasonography D-21-35
Treatment of open animals	Prostaglandin and repeated AI	Ovsynch + repeated AI	Ovsynch + repeated AI

**Table 2 vetsci-11-00218-t002:** Mean values of the fertility parameters of quartiles of cows grouped according to their milk yield.

Quartiles	Milk Yield (kg)	Service Period (Days) ^1^	AI Index ^2^
Group 1	2442–9828	146.1	2.78
Group 2	9829–11,444	158.4	3.10
Group 3	11,445–13,016	170.5	3.37
Group 4	13,017–22,511	187.7	3.77

^1^ Days from calving to conception (open days). All pairwise differences between means are significant after adjustment for multiplicity. ^2^ Number of inseminations per conception. All pairwise differences between means are significant after adjustment for multiplicity.

## Data Availability

The data presented in this study are contained in this article.

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
