# Peer review of "Relationship between Milk Yield and Reproductive Parameters on Three Hungarian Dairy Farms"

_vetsci, 2024, doi:10.3390/vetsci11050218_

Round 1
Reviewer 1 Report
Comments and Suggestions for Authors
General comments
The study investigate the relationship between milk production and fertility in three Hungarian dairy farms, and indicated that there is a significant correlations between milk yield and reproductive parameters -- The number of inseminations per conception (AI index). However, the experiment ignores other external factors that could impact the milk yield and AI index such as body weight, age, breeds, etc. In addition, the contents of the manuscript is more like a communication rather than a research article.
Major issues
1. The introduction is too long. The introduction should be a summary of the background, research condition, and the aim of the paper. It looks like a review.
2. The milk yield and AI index could also be affected by other external factors such as genetic factors, nutrition, body weight, exercise, etc. Without providing the data, it is hard to conclude that the milk yield is only correlated with AI index.
Minor issues
Line 75 and 80. The format of the citation is not correct.
Author Response
Dear Reviewer,
please find attached our reply to your comments.
Many thanks for your efforts.

Reviewer 2 Report
Comments and Suggestions for Authors
In the vetsci-2955824, the authors used more than ten years of data from three farms to clarify the relationship between milk yield and reproductive performance. I thought that analyzing data collected from large dairy farms would be a suitable method for evaluating things such as the number of artificial inseminations and service period. The introduction and discussion cite many papers and scientifically describe the purpose of the research and the interpretation of the results. I thought this paper was well written, except for the following points.
I thought it would be easier to read if the introduction was a little more concise.
In Table 1, Figures 4, and 5, it is necessary for scientific papers to compare data using statistically significant differences.
Since the authors also compare differences between farms, I thought it would be easier to understand if farm characteristics were summarized in a table, not just L217-241.
Reviewer 3 Report
Comments and Suggestions for Authors
· The authors put a lot of effort into it, but unfortunately did not achieve the standard of a good scientific work. The main problem is that the data comes from practical farms. That is, it is possible not to obtain concrete data. This has shown in the results that the data collected is not correct or the statistical analysis is not correct. If the AI index increases, this means that insemination needs to be repeated and the duration of service period extended. This situation clearly occurred in Farm 1, but not in Farms 2 and 3, as shown in Figure 3.
· The number of the references in the text must appear immediately after the reference and not at the end of the paragraph.
· The reference sources in lines between 68 and 80 must be written at the end of the work.
· In the introduction it would have been better if the authors had mentioned the hormonal connections between prolactin and oestrogen and their influence on milk production and reproduction.
· In line 245, the authors mentioned that the milk yield was adjusted for 305 days. The question is how to calculate the milk yield of cows that have given milk for more than 305 days and cows that have given milk for less than 305 days.
· There are contradictions in Figures 4 and 5. In Figure 4 and Farm 3, the AI index was shown higher compared to Farms 1 and 2. But in figure 5 shows the longest service period in Farm 2. This means that there are problems with raw data or errors in statistical analysis and such things need to be fixed.
· The typo in lines 275 and 276 needs to be corrected. Instead of “which” there is “wich”.
· In Figure 2, no unit was specified on the Y-axis.
· Online 372, the word “Recently” was written using other design fonts.
· In the references list, as in the other references, the publication date should be written in bold in lines 477, 538 and 564.
· On line 530, reference number 38, it is not clear whether this is a report or a book, and there is no date, page number, or siren number.
Author Response
Dear Reviewer,
please find attached our reply to your valuable comments.
Many thanks for your efforts.

Reviewer 4 Report
Comments and Suggestions for Authors
Relationship between Milk Yield and Reproductive Parameters 2 in Three Hungarian Dairy Farms
There are keywords simultaneously in the title. These must appear in only one of these locations.
Line 75: It is correctly? “…This value has increased in the following years, reaching 8152 l per cow per year in 2020/21. …..”
The introduction is too long. This should contain a maximum of 2 pages
Need to rewrite the entire 2.2 part. Examined Farms and Technology. There is repeated information that can be written only once.
Line 230: Non pregnant cows were treated with Ovsynch. Animals in the producing deep litter barns
In line 235: “This method is known the best for recognizing the open
cows. The non-pregnant animals were treated by Ovsynch and inseminated repeatedly”
What R packages were used to perform the statistical analyses? Could you show the models used?
Need to improve the quality of images - figures. They are not clear.
It would be interesting to relate this topic - 4.1. Negative Energy Balance (NEB) - with a table showing this relationship Show this from own work: Pregnancy loss may occur at any point of gestation with the largest percentage of loss occurring in the first 30 days and, subsequently, decreasing as the pregnancy progresses.
The authors used 17 citations in the review. All the others appear a posteriori. This should be reviewed, so that the introduction is efficient and functional.
Author Response

(The authors gave the same response as above.)

Round 2
Reviewer 1 Report
Comments and Suggestions for Authors
The authors have answered all my questions.